# Cellular Morphology and Transcriptome Comparative Analysis of *Astragalus membranaceus* Bunge Sprouts Cultured In Vitro under Different LED Light

**DOI:** 10.3390/plants12091914

**Published:** 2023-05-08

**Authors:** Ji Won Seo, Jae Geun Lee, Ji Hye Yoo, Jung Dae Lim, Ik Young Choi, Myong Jo Kim, Chang Yeon Yu, Eun Soo Seong

**Affiliations:** 1Interdisciplinary Program in Smart Science, Kangwon National University, Chuncheon 24341, Republic of Korea; 2Research Institute of Biotechnology, Hwajin Cosmetics, Hongcheon 25142, Republic of Korea; 3Bioherb Research Institute, Kangwon National University, Chuncheon 24341, Republic of Korea; 4Bio-Health Convergence, Kangwon National University, Chuncheon 24341, Republic of Korea; 5Department of Agriculture and Life Industry, Kangwon National University, Chuncheon 24341, Republic of Korea; 6Division of Bioresource Sciences, Department of Applied Plant Sciences, Kangwon National University, Chuncheon 24341, Republic of Korea

**Keywords:** *A. membranaceus* spouts, GO term, LED light sources, transcriptomes, upregulated genes

## Abstract

*Astragalus membranaceus*, the major components of which are saponins, flavonoids, and polysaccharides, has been established to have excellent pharmacological activity. After ginseng, it is the second most used medicinal plant. To examine the utility of *A. membranaceus* as a sprout crop for plant factory cultivation, we sought to establish a functional substance control model by comparing the transcriptomes of sprouts grown in sterile, in vitro culture using LED light sources. Having sown the seeds of *A. membranaceus*, these were exposed to white LED light (continuous spectrum), red LED light (632 nm, 1.58 μmol/m^2^/s), or blue LED light (465 nm, 1.44 μmol/m^2^/s) and grown for 6 weeks; after which, the samples were collected for transcriptome analysis. Scanning electron microscopy analysis of cell morphology in plants exposed to the three light sources revealed that leaf cell size was largest in those plants exposed to red light, where the thickest stem was observed in plants exposed to white light. The total number of genes in *A. membranaceus* spouts determined via de novo assembly was 45,667. Analysis of differentially expressed genes revealed that for the comparisons of blue LED vs. red LED, blue LED vs. white LED, and red LED vs. white LED, the numbers of upregulated genes were 132, 148, and 144, respectively. Binding, DNA integration, transport, phosphorylation, DNA biosynthetic process, membrane, and plant-type secondary cell wall biogenesis were the most enriched in the comparative analysis of blue LED vs. red LED, whereas Binding, RNA-templated DNA biosynthetic process, DNA metabolic process, and DNA integration were the most enriched in the comparative analysis of blue vs. white LED, and DNA integration and resolution of meiotic recombination intermediates were the most enrichment in the comparison between red LED vs. white LED. The GO term associated with flavonoid biosynthesis, implying the functionality of *A. membranaceus*, was the flavonoid biosynthetic process, which was enriched in the white LED vs. red LED comparison. The findings of this study thus indicate that different LED light sources can differentially influence the transcriptome expression pattern of *A. membranaceus* sprouts, which can provide a basis for establishing a flavonoid biosynthesis regulation model and thus, the cultivation of high-functional *Astragalus* sprouts.

## 1. Introduction

*Astragalus membranaceus* Bunge, commonly referred to as Mongolian milkvetch, is an herbaceous perennial plant belonging to the Leguminosae family. In Republic of Korea, high-quality *A. membranaceus* is grown in Jeongseon, Gangwon-do, and has been used as a medicinal plant since ancient times [1,2]. *A. membranaceus* is naturally widely distributed in Asian regions such as Korea and China and in parts of Europe and Africa [3]. In oriental medicine, it is used for diuresis, tonicity, and blood pressure lowering and has also been reported to have blood sugar regulation, antitumor, and antiviral activities [4]. A representative component contained in *A. membranaceus* is formononetin, an isoflavone glycoside, which as a phytoestrogen is established to be a natural substitute for female hormones [5]. In addition, saponins, flavonoids, amino acids, trace elements, and polysaccharides have been reported as major biologically active constituents of *A. membranaceus* [6].

Consequently, environmental damage exacerbated by ongoing climate change and plant factories is increasingly gaining attention as an alternative approach for cultivating high-quality crops in a fixed environment without being influenced by the external environment [7]. Although compared with traditional field cultivation, there are certain disadvantages associated with factory-based cultivation, notably the higher equipment and maintenance costs; various studies are being conducted with the aim of establishing systems that compensate for these disadvantages [8]. Cultivation under plant factory conditions is based on the mechanical regulation of light, humidity, and temperature, and in this regard, there are an increasing number of studies focusing on the use of light-emitting diodes (LEDs) [9]. LEDs are mercury-free, environmentally friendly, and lightweight sources of light that are energy efficient, have a long lifespan, and have advantages such as simple circuitry and the provision of light of a specific quality [10]. LED light can be modified to produce light of different qualities using the principle that light is generated as electron flows, and the light quality can be selected according to the user’s needs [11]. As such, the use of LEDs with specific light quality has been assessed with respect to the cultivation of a range of vegetable crops, although it has rarely been assessed for the cultivation of medicinal plants.

Although the basic data relating to the utilization of LEDs for plant cultivation is gradually accumulating, information pertaining to the identification and control of the relationships between LED light sources and the germination, growth, and contents of functional substances is still far from complete. To facilitate the bulk production of functional substances in crop plants based on the control of LED lighting, it will be necessary to investigate the influence of these light sources on the pattern of genes associated with functional component biosynthesis [12]. Based on this biosynthesis, it would be possible to modify different biological activities and thus, the contents of the functional substances of crops. In this regard, next-generation sequencing (NGS) technology can be applied to obtain large-capacity fragment sequencing data more rapidly and at a lower cost compared with traditional sequencing methods. NGS-based RNA sequencing (high-throughput mRNA sequencing, RNA-seq) can be used to sequence transcriptomic cDNA and quantify the relative amounts of transcript expression. It can be used to identify genes of interest that are specifically expressed according to research objectives, and offers the potential to analyze gene expression, even in the absence of relevant reference genome information [13].

Transcriptomic analysis is a particularly important approach for the comparison of biological activities and differences in gene expression in different plant tissues and for elucidating genome functionality, and numerous high-quality new crop varieties have been developed based on the genetic improvement of the plant’s secondary metabolites using genetic engineering technology. Moreover, advances in transcriptomics have made it possible to identify genes associated with specific functions and predict hitherto unknown genes. Recently, the scope of this research expanded to include medicinal plants, for which comparative transcriptome analyses have been successfully performed [14,15]. To complement this growing body of research, we sought in the present study to analyze the transcriptomic responses of *A. membranaceus* sprouts grown under in vitro conditions in which plants were exposed to LED light sources of different wavelengths. This comparative transcriptome analysis enabled us to establish a light environmental control model that could be applied to enhance the production of functional substances of interest in sprouts of the medicinal plant *A. membranaceus*.

## 2. Results

### 2.1. Comparison of the Cellular Morphologies of A. membranaceus Sprouts Germinated In Vitro under Three Different LED Light Sources

To compare the tissue morphologies of in vitro-cultured *A. membranaceus* sprouts exposed to different light sources for 6 weeks, we performed scanning electron microscopy observations of leaf and stem cross-sections (Figure 1). Among the different treatments, the largest leaf cells (58.57 ± 6.17 μm) were observed in plants exposed to the red LED, whereas plants illuminated with white LED light were found to have the smallest leaf cells (20.67 ± 6.50 μm) by cross-sections. With regards to stem cell width, the highest and lowest values of 22.17 ± 2.44 and 16.81 ± 1.89 μm were observed in plants exposed to white and red LED lights, respectively (Table 1).

### 2.2. RNA-Seq of A. membranaceus Sprouts Germinated In Vitro under Three Different Light Sources

To compare the gene expression profiles of in vitro-germinated *A. membranaceus* sprouts exposed to different light sources for 6 weeks, we performed transcriptome analysis. In terms of raw RNA-seq data, we obtained 38,343,876; 41,164,444; and 33,223,692 reads for plants cultivated under blue, red, and white LEDs, respectively, following appropriate trimming, we obtained 36,722,118 (93.97%), 39,275,548 (93.34%), and 31,703,398 (92.01%) clean reads, respectively (Table 2). Based on de novo assembly, we detected a total of 45,667 genes in the germinated sprouts.

### 2.3. Analysis of Differentially Expressed Genes in A. membranaceus Sprouts Germinated In Vitro under Three Different Light Sources

Table 3 shows the mapping results, which were applied to analyze the genes expressed in *A. membranaceus* exposed to the three different light sources, using the 45,667 genes identified via de novo assembly as a reference. For plants cultivated under blue, red, and white LED lights, we obtained 25,740,782 (70.1%), 26,850,570 (68.4%), and 20,898,430 (65.9%) mapped reads, respectively, with an average of 68.1% of the data being successfully mapped to the reference genome. Based on the confirmation of gene expression levels via differentially expressed gene (DEG) analysis, we established the numbers of genes to which at least one clean read was mapped for each of the 45,667 total genes, with 42,572; 42,542; and 42,606 clean data being mapped for plants exposed to blue, red, and white LED light, respectively (Table 4). Using the criterion of a log_2_-fold change, we established that the number of up- and downregulated genes between blue LED and red LED, blue LED and white LED, and red light and white LED treatment groups were 132 and 153, 148 and 93, and 144 and 91, respectively (Table 5). Figure 2 shows an MA Plot generated using the data presented in Table 5, in which genes showing significant differences in expression between two samples are indicated by a red color. To confirm correlations between the two samples, we generated a correlation plot, on the basis of which we determined that the highest correlation between different treatments (0.94) was obtained for *A. membranaceus* germinated in vitro under white and blue LED light (Figure 2).

### 2.4. GO Analysis of A. membranaceus Sprouts Germinated In Vitro under Three Different Light Sources

The results of GO analysis of up and downregulated genes revealed a significant difference in expression level when comparing the two samples based on DEG analysis (Figure 3 and Figure 4). When the same treatment group was compared, cellular process, catalytic activity, and cellular process showed the highest expression levels with 51, 50, and 56 genes among the upregulated genes, respectively. For blue LED vs. red LED and blue LED vs. white LED comparisons, the largest proportions of downregulated genes were found to be involved in catalytic activity (68 and 32 genes, respectively). For the comparison of red LED and white LED treatment groups, we found that 33 genes associated with cellular processes were the most downregulated genes. A comparison of the expression levels of up- and downregulated genes in the different GO functional categories revealed that in the molecular function category, genes associated with binding and catalytic activity were characterized by the highest levels of expression. In the cellular component category, except for the downregulated genes between the blue LED and white LED treatment groups, we detected high levels of expression for all cellular anatomical entity-related genes. With respect to the biological process category, we detected generally high expression levels among genes associated with cellular processes. Expression levels of metabolic process-related genes were also high, with the exception of the downregulated genes between blue LED and white LED treatment groups and upregulated genes between the red LED and white LED groups.

### 2.5. qPCR Analysis for Reference Genes by GO Analysis

To confirm the reliability of the transcriptome comparative data, qPCR was performed by selecting five genes upregulated in white LED light and five genes upregulated in red LED light based on the blue LED light (Figure 5). As a result of the qPCR analysis, it was confirmed that the expression patterns of all genes upregulated in the white and red light sources were represented by correlating in the transcriptome analysis. In particular, among the 10 genes, it was confirmed that the expression levels of the reference genes Gene_174710T and Gene_334410T, which are upregulated in the white LED light source, were significantly higher than those of other LED light sources.

### 2.6. Up- and Downregulation of Transcriptomes in A. membranaceus Sprouts Treated with Blue LED Light vs. Red LED Light

The GO term with the highest number of upregulated transcriptomes was binding (GO:0005488), for which 12 were counted. The GO terms with the next highest number of upregulated transcriptomes (six) were DNA integration and transport. The GO terms with five counted transcriptomes were phosphorylation, DNA biosynthetic process, membrane, and plant-type secondary cell wall biogenesis. There were 12 transcriptomes counted as two and 28 transcriptomes counted as one (Table 6). Among the downregulated transcriptomes, the most represented GO term was binding (GO:0005506) with eight counts, followed by DNA integration (GO:0015074) with seven counts. The five-count GO term was the integral component of the membrane, and the four-count GO term was carbon utilization. The three counts were obtained for telomere maintenance (GO:0000723), defense response (GO:0006952), regulation of DNA-templated transcription (GO:0006355), protein ubiquitination (GO:0016567), and methylation (GO:0032259). Two counts were classified as GO terms for 11 transcriptomes, and one count was classified as the GO term for 50 transcriptomes (Table 7).

### 2.7. Up- and Downregulation of Transcriptomes in A. membranaceus Sprouts Treated with Blue LED Light vs. White LED Light

Among the transcriptome classified as upregulated, the most represented GO term was Binding (GO:0005488) with eight counts. RNA-templated DNA biosynthetic processes were classified as a six-count transcriptome, and five counts were obtained for DNA metabolic processes (GO:0006259) and DNA integration (GO:0015074). Membrane (GO:0016020, proteolysis (GO:0006508), and plant-type secondary cell wall biogenesis (GO:0009834) were represented by four counts. Three-count terms were fatty acid biosynthetic process (GO:0006633) and glycosyltransferase activity (GO:0016757). Eight of the two-count transcriptomes and 45 of 1-count transcriptomes were upregulated under blue LED-blue vs. white LED light (Table 8). A total of 42 transcriptomes for GO terms were downregulated. Five-count transcriptomes were obtained for DNA integration (GO:0015074) and binding (GO:0005488). The three-count GO terms included the inositol catabolic process (GO:0019310), RNA-templated DNA biosynthetic process (GO:0006278), and translational initiation (GO:0006413). Two counts were classified into four GO terms, and one count was classified into 33 GO terms (Table 9).

### 2.8. Up- and Downregulation of Transcriptomes in A. membranaceus Sprouts Treated with Red LED-Light vs. White LED-Light

Transcriptomes showing upregulation in red LED vs. white LED comparisons appeared in all 60 GO terms. Among these, the highest count (seven) was obtained for DNA integration (GO:0015074) term. Next, the resolution of meiotic recombination intermediates (GO: 0000712) was represented by six counts. Five counts were classified as proteolysis (GO:0006508), and four counts were grouped as GO terms of flavonoid biosynthetic process (GO:0009813) and defense response (GO:0006952). GO terms with three counts were signal transduction (GO: 0007165), nucleic acid metabolic process (GO: 0090304), and xyloglucan metabolic process (GO: 0010411). In addition, two counts were classified as upregulation of 10 GO terms and one count of 42 GO terms (Table 10). In the downregulation transcriptome result of red LED vs. white LED light, DNA integration (GO:0015074) was represented by six counts. The three-count GO terms were nucleic acid metabolic process (GO:0090304) and binding (GO:0005488), whereas two-count transcriptomes were classified into five GO terms, and one-count transcriptomes were classified into 35 GO terms (Table 11).

## 3. Discussion

The quality of light influences the activity of multiple biological pathways in plants and can accordingly have a significant effect on morphological phenotypes. To date, however, there appear to have been no reports regarding morphological changes at the cellular level or a comparative analysis of transcripts in aseptically cultivated plants of the medicinal plant *A. membranaceus* exposed to different colored LED lights. Previous studies on *A. membranaceus* transcriptomes have tended to focus on plants grown in general classical cultivation environments [16,17]. In the present study, we analyzed and compared the morphological changes and metabolic mechanisms at the transcriptomic level in *A. membranaceus* sprouts germinated in vitro under aseptic conditions and illuminated with LED lights of three different wavelengths. The transcriptomic data obtained from this comparative analysis will contribute to gaining a better understanding of the molecular mechanisms underlying the response of *A. membranaceus* to LED light of different colors, and thereby provide a basis for cultivating medicinal plants with enhanced functional properties.

At the cellular level, we found the in vitro-germinated *A. membranaceus* sprouts that had been exposed to red LED light were characterized by the largest leaf cells, whereas the thickest stem cells were observed in plants cultivated under white LED illumination (Table 1). Previous studies on the effects of different LEDs on *Astragalus* growth have found that the growth of *A. membranaceus* plants cultivated under blue LED illumination was superior to that of plants exposed to either white LED or red LED light when visually measured [18], although these authors detected no significant differences among plants treated with these three light sources with respect to the length and number of leaves. In studies on the effects of LED illumination on the growth of other plants, it has been shown that in *Myrtus communis*, the highest shoot multiplication effect occurred in plants treated with 5 µM BA(Benzyladenine) grown under red LED light [19], whereas, in *Salvia miltiorrhiza*, Choi et al. (2020) found red LED light to be more effective than blue LEDs in promoting leaf length and width [20]. Among other studies, it has been reported that blue LED light inhibits vegetative growth, such as seedling growth and root formation, whereas green LED light has been established to have negative effects on plant growth and fresh and dry weights [21,22]. However, although many studies have investigated differences in plant growth responses using LED lights, most assessments of differences in growth have been based on unaided visual evaluations, whereas relatively few studies have examined responses at the cellular level. In this study, using scanning electron microscopy, we examined the leaf cell size and stem thickness of *in vitro*-germinated *A. membranaceus* sprouts and accordingly demonstrated red and white LED light sources to be the most effective in promoting leaf cell expansion and stem thickening, respectively. Notably, in this context, the findings of previous studies have indicated that the responses of plants to light of different qualities tend to differ among species. Accordingly, we speculate that our observations for *A. membranaceus* might also apply to other species in the family Leguminosae. In addition, it has also been reported that exposure to white LED light enhances the antioxidant activity of *A. membranaceus* [18]. Thus, we assume that the thickening of the stems of *A. membranaceus* grown under white LED light contributes to enhancing stress resistance and antioxidant activity.

Although there have been several transcriptome analyses using *A. membranaceus*, the present study is the first to analyze the transcriptome of *A. membranaceus* sprouts germinated in vitro under sterile conditions. Comparative analysis of plants treated with the three LED light sources revealed that the expression levels of upregulated transcripts in plants exposed to white LED were higher than those in plants cultivated under either red or blue LED sources, indicating that the number of downregulated transcripts is also lowest under white LED light illumination. To functionally annotate the DEGs between different light source treatments, we performed GO analysis. For the blue LED vs. red LED comparison, we detected seven enriched GO sub-categories (phosphorylation, DNA biosynthetic process, binding, membrane, DNA integration, and transport). Similarly, seven enriched categories were detected for the blue LED vs. white LED comparison (binding, RNA-templated DNA biosynthetic process, DNA metabolic process, DNA integration, membrane, proteolysis, and plant-type secondary cell wall biogenesis), whereas for the red LED vs. white LED comparison, transcripts were classified into five categories (DNA integration, resolution of meiotic recombination intermediates, proteolysis, flavonoid biosynthetic process, and defense response). 

Conversely, with respect to light source-specific downregulated transcripts, we detected enrichment of DNA integration, an integral component of membrane, and carbon utilization for the blue LED vs. red LED comparison; enrichment of DNA integration and binding for the blue LED vs. white LED comparison; and enrichment of DNA integration for the red LED vs. white LED comparison. Among the upregulated transcripts between blue LED- and red LED-illuminated plants, the most highly enriched GO category was binding (GO:0005488). A GO category that was commonly enriched with upregulated transcripts in all three comparisons is DNA integration (GO:0015074), which is known to function in biological processes that involve the integration of DNA segments into other larger DNA molecules, such as chromosomes. Similarly, the DNA integration category was found to be enriched with downregulated transcripts. We accordingly speculate that each of the three LED light sources has a significant effect on plant cell size and morphogenesis by directly influencing biological processes in sterile *A. membranaceus* sprouts.

Light plays a particularly important role in the success of in vitro plant tissue culture. LEDs of many artificial light sources are important for plant mass propagation systems in which the color of the light source affects plant growth and development. It has been established that the color of LED light influences the morphogenesis, differentiation, and proliferation of plant cells, tissues, and organ cultures and is essential for the production of secondary metabolites [23,24]. Among the diverse spectrum of secondary metabolites, phenolic compounds have been established to have a broad range of biological properties, including antioxidant, anticancer, antibacterial, and antiallergic activities. As phenolic compounds, flavonoids have been found to have antioxidant and anti-inflammatory properties and play roles in the inhibition of cell division and redox regulation of cells [25,26]. In this regard, an enrichment of phenolic components and high antioxidant capacity has been reported in the leaves and roots of *S. miltiorrhiza* plants exposed to LED light sources [20]. Furthermore, analysis of phenolic contents in the leaves and roots of *Rehmannia glutinosa* has revealed that total phenolic contents were highest in those plants exposed to blue LED light, although the highest flavonoid contents were detected in plant cultivated under red LED illumination [27,28]. Changes in the activity of plants promoted by exposure to LED light sources are mediated via the activities of associated genes. In this regard, we found that a GO term of particular interest to us, namely, flavonoid biosynthetic process (GO:0009813), was enriched with four genes that were differentially expressed between red LED- and white LED-illuminated plants. Similar enrichment was detected for the defense response term (GO:0006952), thereby indicating a correlation between plant defense and flavonoid biosynthetic mechanisms. 

Flavonoids are a notably abundant class of secondary metabolites present in all terrestrial plants, with more than 10,000 different types believed to occur in different plant species. Flavonoids are a class of phenylpropanoids derived from the shikimate and acetate pathways via the activity of cytoplasmic multienzyme complexes anchored in the endoplasmic reticulum [29]. As defense compounds and developmental regulators, it has been revealed that they can play diverse roles in plant–nematode defense mechanisms by acting as defense compounds or signal molecules that have inhibitory effects on nematodes at different life stages [30]. Therefore, it would be desirable to study the relationships between plant disease defense and various stresses as functions of genes specifically expressed in aseptically cultured *A. membranaceus* sprouts exposed to white LED light. In addition, it is suggested that the light condition for improving the biomass of *A. membranaceus* sprouts requires the appropriate use of white light and red light sources.

## 4. Materials and Methods

### 4.1. Preparation of In Vitro-Cultured A. membranaceus Sprouts under Artificial Light Source Conditions

The seeds of *A. membranaceus* used in this study were purchased from KS Jongmyo (Incheon, Republic of Korea). Seed sterilization was performed to obtain aseptic *A. membranaceus* sprouts. The seeds were initially placed in 70% EtOH and shaken for 1 min, after which they were transferred to 3% NaClO and shaken again for 5 min. Following five washes with sterile distilled water, the sterilized seeds were transferred to culture bottles containing agar-solidified Murashige and Skoog medium and cultured for 6 weeks. For illumination, we used LED lights of different wavelengths, namely, white (continuous spectrum), red (632 nm, 1.58 μmol/m^2^/s), and blue (465 nm, 1.44 μmol/m^2^/s). The wavelengths of these LED lights were measured using a PG200N illuminometer (United Power Research Technology Co., Zhunan Township, Taiwan). The photoperiod to which the plants were exposed was set to 16 h light and 8 h dark, which was maintained for the 6-week growth period (Figure 6).

### 4.2. Scanning Electron Microscope Analysis

Leaf and stem samples collected from *A. membranaceus* plants were fixed with 2% glutaraldehyde and 2% paraformaldehyde in 50 mM cacodylate buffer (pH 7.4) for 1 h at 4 °C [31]. Thereafter, the fixed samples were dehydrated in a graded ethanol series for 10 min and then immersed in mixtures of 100% ethanol and isoamyl acetate (2:1, 1:1, and 1:2), each for 10 min. After immersion in pure isoamyl acetate for 15 min, the isoamyl acetate was removed, and the samples were dried using a critical point dryer. Dried samples were then sputter-coated with a thin layer of gold. Observations were performed at the Korea Basic Science Institute, Chuncheon, using a SUPRA 55VP scanning electron microscope (Carl Zeiss, Oberkochen, Germany) operating at an acceleration voltage of 3 kV. The leaf samples were observed by taking the fourth leaf from the top and measuring 3 leaves.

### 4.3. RNA-Seq Library Construction and Sequencing

Total RNA was extracted from in vitro-cultured *A. membranaceus* sprouts using Trizol reagent (Invitrogen Scientific, Inc., Waltham, MA, USA), with the purity of the extracted RNA being determined using a microvolume spectrophotometer (Keen Innovative Solutions, Daejeon, Republic of Korea). RNA-seq libraries were constructed using a TruSeq RNA kit (Illumina Inc., San Diego, CA, USA) and sequenced using the Illumina HiSeq 2500 platform (Illumina Inc., San Diego, CA, USA).

### 4.4. Analysis of Differential Gene Expression

The Illumina sequencing data were initially pre-processed using Trimmomatic v0.39 (http://www.usadellab.org/cms/) (30 May 2022). Reads of 50 bp or less were removed, as were low-quality reads when the average quality per base was less than 20 bp, determined by applying a 4-base side sliding window. Clean reads were mapped to the reference sequence using HISAT2 v2.1.0 (http://daehwankimlab.github.io/hisat2/) (30 May 2022) and Samtools v1.13 (http://www.htslib.org/) (30 May 2022). The number of mapped reads was confirmed using the program HTSeq v0.11.2 (http://htseq.readthedocs.io/en/master/) (30 May 2022). Thereafter, normalization and analysis of differential gene expression were performed using the DESeq v1.38.0 program (https://www.bioconductor.org/packages//2.10/bioc/html/DESeq.html) (30 May 2022). The differential expression of genes was defined based on log_2_-fold change values, with values greater and less than 2 being taken to be indicative of up- and downregulation, respectively. To identify genes showing significantly different expressions, we prepared MA plots based on log_2_-fold change and q-values, and correlations between the two samples were shown using correlation plots.

### 4.5. qPCR Analysis of Reference Genes

cDNA was synthesized using PrimeScript™ RT Master Mix (Perfect Real Time) (Takara Korea Biomedical Inc., Seoul, Republic of Korea) with total RNA used for transcriptome analysis. The qPCR reaction was performed with a CronoSTAR™ 96 Real-Time PCR System (Takara Korea Biomedical Inc., Seoul, Republic of Korea) using TOPreal™ SYBR Green qPCR PreMIX (Enzynomics Co., Ltd., Daejeon, Republic of Korea). qPCR conditions were performed by initial denaturation at 95 °C for 10 min with a volume of 25 μL, followed by three-step amplification (denaturation 95 °C 10 s, annealing 60 °C 15 s, elongation 72 °C 15 s) at 45 cycles. The primers used for qPCR were designed in Primer3Plus, and the nucleotide sequences are shown in Table 12.

### 4.6. Gene Annotation and Functional Analysis

For the purposes of Gene Ontology (GO) analysis, following an initial BLAST (v2.12.0+: ftp://ftp.ncbi.nlm.nih.gov/blast/executables/blast+/LATEST/, accessed on 30 May 2022) search of the NCBI database and implementation of the EMBL-EBI InterProScan program v5.56-89.0 (https://www.ebi.ac.uk/interpro/search/sequence/, 30 May 2022), the data obtained were integrated using the BLAST2GO program v6.0.3 (https://www.blast2go.com/, 30 May 2022) to confirm GO analysis and annotation results. GO analysis involved the classification of genes into three categories molecular function, cellular component, and biological process.

## 5. Conclusions

Our comparative analysis of the transcriptomes of in vitro-germinated *A. membranaceus* sprouts exposed to artificial lighting of three different colors revealed differences in the leaf cell size and stem cell thickness of plants cultured under different light sources. RNA-sequencing of samples obtained from plants exposed to the different light sources yielded 38,343,876; 41,164,444; and 33,223,692 raw reads for those plants exposed to blue, red, and white LEDs, respectively. A total of 45,667 genes expressed in *A. membranaceus* sprouts were analyzed based on de novo assembly. Upregulated transcripts associated with flavonoid biosynthesis-related mechanisms were detected in plants treated with white LED light. Given that these results were obtained at the whole-plant level, it would be desirable in future studies to determine the expression profiles of secondary metabolites in different plant tissues. Nevertheless, based on our findings in this study, we anticipate further development of the plant factory cultivation of high-functional medicinal plants as sprout vegetables, thereby enhancing their value.

## Figures and Tables

**Figure 1 plants-12-01914-f001:**
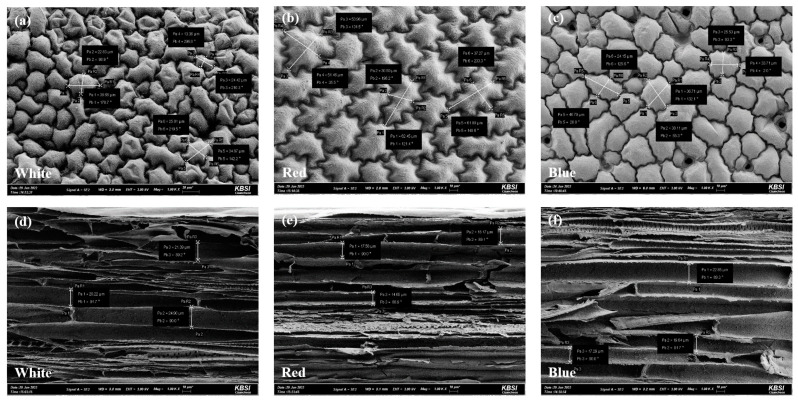
Cellular morphology analysis using SEM image of *A. membranaceus* sprouts grown in vitro cultured under three types of LED lights for 6 weeks. ((**a**–**c**): leaf, (**d**–**f**): stem).

**Figure 2 plants-12-01914-f002:**
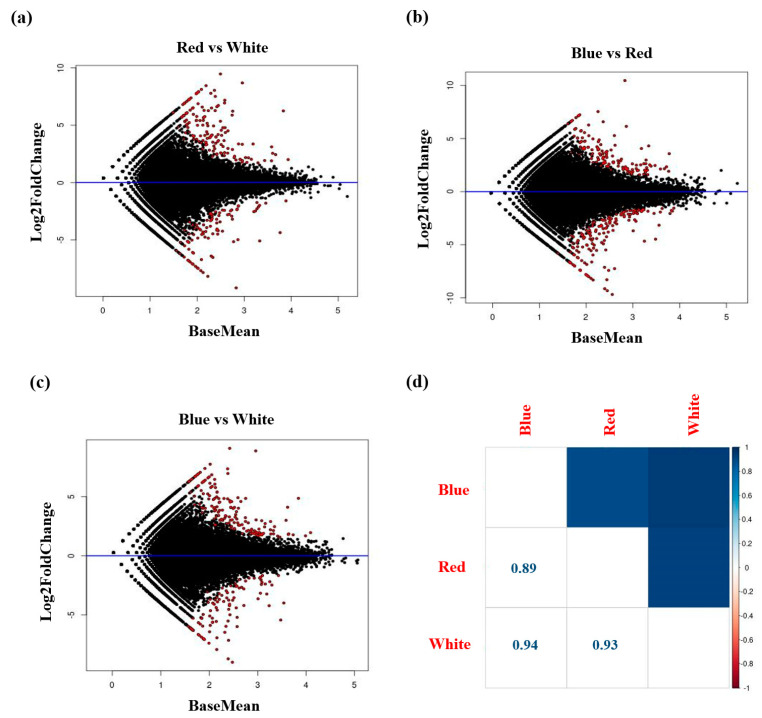
(**a**–**c**) Examples of differentially expressed genes (red points) identified by the MA-plot-based method of *A. membranaceus* sprouts under three types of LED lights. (**d**) A matrix showing the correlation for DEGs analysis of *A. membranaceus* sprouts under three types of LED lights.

**Figure 3 plants-12-01914-f003:**
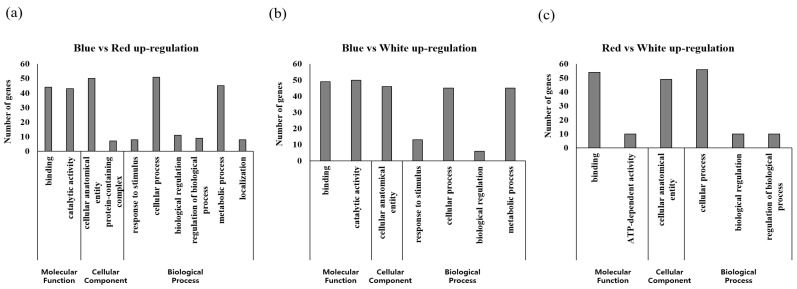
Upregulated genes through Gene Ontology (GO) analysis of differentially expressed genes (DEGs) in *A. membranaceus* sprouts under three types of LED light sources. (**a**) Blue vs. Red up-regulated gene expression, (**b**) Blue vs. White up-regulated gene expression, (**c**) Red vs. White up-regulated gene expression.

**Figure 4 plants-12-01914-f004:**
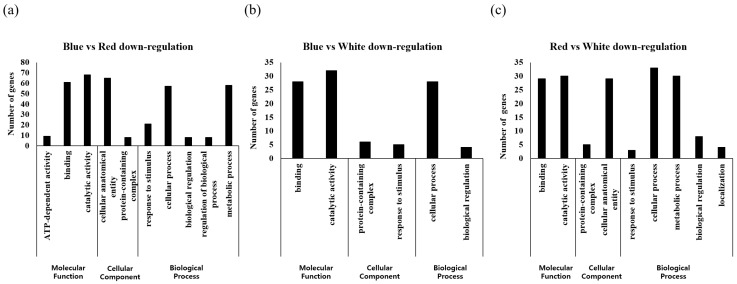
Downregulated genes through Gene Ontology (GO) analysis of differentially expressed genes (DEGs) from *A. membranaceus* sprouts under three types of LED light sources. (**a**) Blue vs. Red down-regulated gene expression, (**b**) Blue vs. White down-regulated gene expression, (**c**) Red vs. White down-regulated gene expression.

**Figure 5 plants-12-01914-f005:**
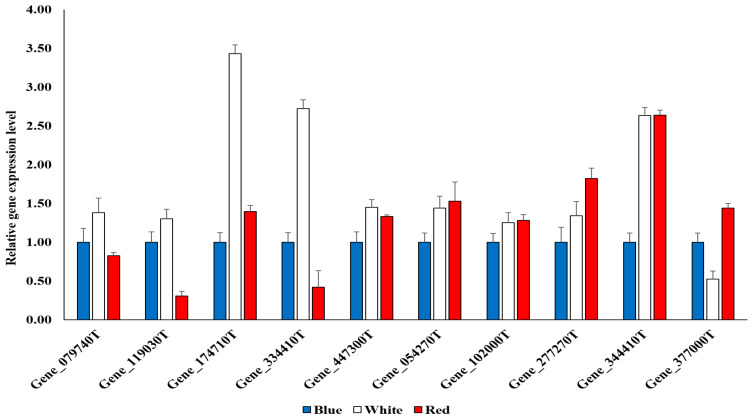
Upregulated reference genes expressed in white LED light (Gene_079740T, Gene_119030T, Gene_174710T, Gene_334410T, and Gene_447300T) and red LED light (Gene_054270T, Gene_102000T, Gene_277270T, Gene_344410T, and Gene_377000T) based on blue LED light through Gene Ontology (GO) analysis of differentially expressed genes (DEGs) of *A. membranaceus* sprouts under three types of LED light sources.

**Figure 6 plants-12-01914-f006:**
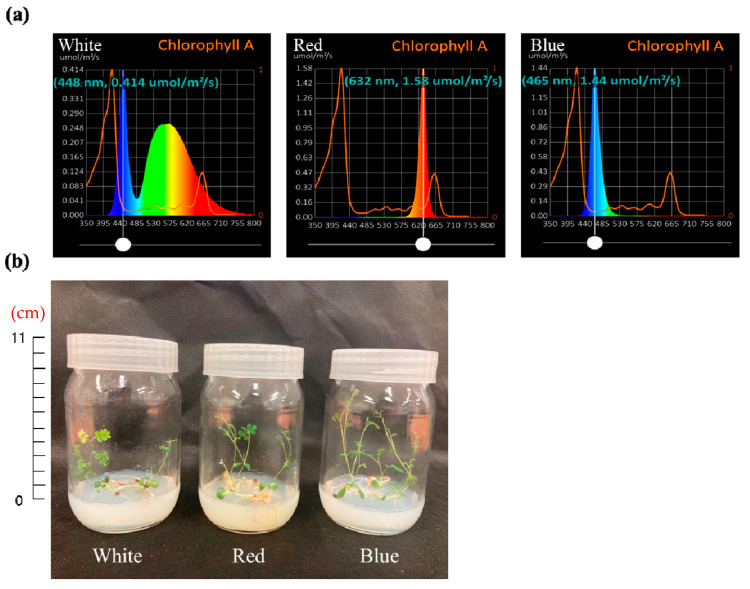
Wavelengths of each LED light source (**a**) and *A. membranaceus* sprouts (**b**) grown in vitro cultured under three types of LED lights for 6 weeks.

**Table 1 plants-12-01914-t001:** Analysis of differences in cell size of leaf and cell thickness of stem using SEM image of *A. membranaceus* sprout under three types of LED light sources.

LED Light Sources ^(1)^	White	Red	Blue
Size of leaf cell (μm)	20.67 ± 6.50 ^c^	58.57 ± 6.17 ^a^	36.18 ± 2.25 ^b^
Thickness of stem cell (μm)	22.17 ± 2.44 ^a^	16.81 ± 1.89 ^b^	19.93 ± 2.79 ^ab^

^(1)^ LED light sources; three different types of LED light sources composed of white (continuous spectrum), red (632 nm, 1.58 μmol/m^2^/s) and blue (465 nm, 1.44 μmol/m^2^/s). Mean values ± SD from triplicate-separated experiments are shown (n = 3). Means within a column followed by the same letter are not significantly different based on the Duncan’s Multiple Range Test (DMRT, *p* < 0.05).

**Table 2 plants-12-01914-t002:** Sequences information of *A. membranaceus* sprouts under three types of LED light sources.

Sample	Raw No	Raw Length	Clean No	Clean Length	Clean %
White	33,223,692	5,016,777,492	31,703,398	4,615,844,328	92.01
Red	41,164,444	6,215,831,044	39,275,548	5,802,087,803	93.34
Blue	38,343,876	5,789,925,276	36,722,118	5,440,840,126	93.97

**Table 3 plants-12-01914-t003:** Mapping result of aseptic cultured *Astragalus membranaceus* under different types of artificial light sources (LEDs).

Sample Name	White	Red	Blue
Total Reads No	31,703,398	39,275,548	36,722,118
Mapped PE Reads No	20,898,430	26,850,570	25,740,782
% Mapped PE Reads No	65.9	68.4	70.1

**Table 4 plants-12-01914-t004:** Expressed gene number of aseptic cultured *Astragalus membranaceus* under different types of artificial light sources (LED).

Sample Name	White	Red	Blue
0	3061	3125	3095
>0	42,606	42,542	42,572

**Table 5 plants-12-01914-t005:** Up- and down-expressed gene number of aseptic cultured *Astragalus membranaceus* under different types of artificial light sources (LEDs).

Sample Name	Blue vs. Red	Blue vs. White	Red vs. White
Up	132	148	144
Down	153	93	91

**Table 6 plants-12-01914-t006:** GO enrichment of upregulated transcripts in blue LED vs. red LED light of *A. membranaceus* sprouts (*p*-value < 0.05).

GO Terms	GO ID	Count	*p*-Value
Phosphorylation	GO:0046777	5	0.0001143
DNA biosynthetic process	GO:0006278	5	6.78 × 10^−10^
Binding	GO:0005488	12	4.32 × 10^−11^
Membrane	GO:0016020	5	0.000363
DNA integration	GO:0015074	6	3.69 × 10^−5^
Transport	GO:0015918	6	2.8 × 10^−7^
Plant-type secondary cell wall biogenesis	GO:0009834	5	6.67 × 10^−8^
Proteolysis	GO:0006508	2	9.68 × 10^−6^
DNA replication	GO:0006260	2	4.26 × 10^−5^
Nucleotide-excision repair	GO:0006289	2	3.16 × 10^−5^
Regulation of DNA-templated transcription	GO:0006355	2	1.19 × 10^−5^
Negative regulation of endopeptidase activity	GO:0010951	2	1.6 × 10^−7^
DNA metabolic process	GO:0006259	2	0.000323
Plant-type primary cell wall biogenesis	GO:0009833	2	0.000204
UDP-glycosyltransferase activity	GO:0008194	2	0.000103
Regulation of catalytic activity	GO:0043086	2	1.98 × 10^−7^
Phloem development	GO:0010088	2	5.33 × 10^−8^
Fatty acid biosynthetic process	GO:0006633	2	1.98 × 10^−5^
Nucleic acid metabolic process	GO:0090304	2	0.000201
Regulation of translation	GO:0006417	1	0.000227
UDP-rhamnose biosynthetic process	GO:0010253	1	1.36 × 10^−5^
Proteolysis involved in protein catalytic process	GO:0051603	1	0.000219
Signal peptide processing	GO:0006465	1	6.34 × 10^−5^
Response to light stimulus	GO:0009416	1	4.57 × 10^−5^
Organelle organization	Go:0006996	1	3.24 × 10^−6^
Cellulose microfibril organization	GO:0010215	1	0.000328
Arginyl-tRNA aminoacylation	GO:0006420	1	0.000249
Exonucleolytic trimming to generate mature 3’-end of 5.8S rRNA from tricistronic rRNA transcript (SSU-rRNA, 5.8S rRNA, LSU-rRNA)	GO:0000467	1	0.000278
Xylan catalytic process	GO:0008194	1	1.14 × 10^−5^
RNA phosphodiester bond hydrolysis	GO:0090502	1	0.000142
Positive regulation of GTPase activity	GO:0043547	1	1.95 × 10^−8^
Modulation of process of another organism	GO:0035821	1	2.93 × 10^−5^
Sucrose biosynthetic process	GO:0010088	1	2.1 × 10^−5^
Response to water deprivation	GO:0009414	1	5.51 × 10^−8^
Shikimate O-hydroxycinnamoyltransferase activity	GO:0047172	1	0.000372
Cell cycle	GO:0007049	1	1.52 × 10^−5^
Double-strand break repair via homologous recombination	GO:0000724	1	8.6 × 10^−5^
Sterol biosynthetic process	GO:0016126	1	3.99 × 10^−6^
Chloroplast rRNA processing	GO:1901259	1	2.59 × 10^−6^
Microtubule-based movement	GO:0007018	1	3.79 × 10^−5^
Inositol catalic process	GO:0019310	1	7.33 × 10^−6^
Group II intron splicing	GO:0000373	1	2.82 × 10^−6^
Autophagosome assembly	GO:0000045	1	1.86 × 10^−5^
Defense response	GO:0006952	1	7.72 × 10^−7^
transcription initiation at RNA polymerase II promoter	GO:0006367	1	0.000564
Inositol biosynthetic process	GO:0006021	1	0.000497
Negative regulation of translation	GO:0017148	1	0.000489

**Table 7 plants-12-01914-t007:** GO enrichment of downregulated transcripts in blue LED vs. red LED light of *A. membranaceus* sprouts (*p*-value < 0.05).

GO Terms	GO ID	Count	*p*-Value
Binding	GO:0005506	8	6.14 × 10^−5^
DNA integration	GO:0015074	7	2.01 × 10^−11^
Integral component of membrane	GO:0016021	5	0.000251
Carbon utilization	GO:0015976	4	4.06 × 10^−6^
Telomere maintenance	GO:0000723	3	9.85 × 10^−6^
Defense response	GO:0006952	3	1.79 × 10^−5^
Regulation of DNA-templated transcription	GO:0006355	3	4.25 × 10^−7^
Protein ubiquitination	GO:0016567	3	3.97 × 10^−7^
Methylation	GO:0032259	3	5.3 × 10^−5^
Translational initiation	GO:0006413	3	2.64 × 10^−13^
Tranmembrane transport	GO:0055085	2	6.74 × 10^−5^
Response to light stimulus	GO:0009416	2	0.000494
Carbohydrate metabolic process	GO:0005975	2	0.000487
RNA phosphodiester bond hydrolysis, endonucleolytic	GO:0090502	2	1.53 × 10^−7^
DNA-templated DNA replication	GO:0006261	2	0.000119
Oxidoreductase activity	GO:0016491	2	0.000173
Translational elongation	GO:0006414	2	4.12 × 10^−8^
Helicase activity	GO:0004386	2	8.81 × 10^−5^
DNA metabolic process	GO:0006259	2	0.000277
Extracellular space	GO:0005615	2	5.67 × 10^−5^
Microtuble-based movement	GO:0007018	2	0.00024
Glycogenin glucosyltransferase activity	GO:0008466	1	0.000138
Signal transduction	GO:0007165	1	8.13 × 10^−6^
Protein phosphorylation	GO:0006468	1	0.000124
‘de novo’ IMP biosynthetic process	GO:0006189	1	0.000216
Transmembrane phosphate ion transport from cytosol to vacuole	GO:1905011	1	4.24 × 10^−6^
tRNA methylation	GO:0030488	1	1.72 × 10^−14^
Pseudouridine synthesis	GO:0001522	1	0.000239
Micromolecule biosynthetic process	GO:0009059	1	1.4 × 10^−5^
Inositol catalic process	GO:0019310	1	2.07 × 10^−5^
Chloroplast thylakoid membrane	GO:0009535	1	3.39 × 10^−5^
Catalytic activity	GO:0003824	1	1.39 × 10^−11^
Circadian regulation of gene expression	GO:0032922	1	8.33 × 10^−5^
Dephophorylation	GO:0016311	1	0.000119
Gene silencing by RNA-directed DNA methylation	GO:0080188	1	2.9 × 10^−7^
Monooxygenase activity	GO:0004497	1	8.06 × 10^−5^
Biosynthetic process	GO:0009058	1	0.000131
Chlorophyllase activity	GO:0047746	1	1.71 × 10^−5^
4-coumarate-CoA ligase activity	GO:0016207	1	6.53 × 10^−6^
Gluconeogenesis	GO:0006094	1	1.41 × 10^−7^
Nucleic acid binding	GO:0003676	1	1.04 × 10^−9^
Response to oxidative stress	GO:0006979	1	6.37 × 10^−5^
Naringenin 3-dioxygenase activity	GO:0045486	1	1.57 × 10^−7^
Response to oomycetes	GO:0002239	1	1.16 × 10^−7^
Oxidative photosynthetic carbon pathway	GO:0009854	1	7.15 × 10^−19^
Citrate transport	GO:0015746	1	3.5 × 10^−7^
Lipid metabolic process	GO:0006629	1	0.000336
Proteolysis	GO:0006508	1	8.26 × 10^−5^
Chloroplast organization	GO:0009658	1	1.36 × 10^−5^
Photosynthetic electron transport in photosystem I	GO:0009773	1	0.000381
Triglyceride lipase activity	GO:0004806	1	0.000525
Chaperone-mediated protein folding	GO:0061077	1	0.000294
Glycolytic process	GO:0006096	1	4.23 × 10^−6^
Nucleoside metabolic process	GO:0009116	1	3.43 × 10^−7^
Resolution of meiotic recombination intermediates	GO:0000712	1	1.3 × 10^−6^
DNA repair	GO:0006281	1	0.000496
Photoreactive repair	GO:0000719	1	1.08 × 10^−6^
UDP-D-Xylose biosynthetic process	GO:0033320	1	0.000123
Glutathione metabolic process	GO:0006749	1	1.78 × 10^−6^
Amino acid transport	GO:0006865	1	2.09 × 10^−6^
RNA-templated DNA biosynthetic process	GO:0006278	1	3.81 × 10^−5^
Regulation of gene expression	GO:0010468	1	2.58 × 10^−6^
Dioxygenase activity	GO:0051213	1	0.000228
Protein import into nucleus	GO:0006606	1	3.83 × 10^−5^
Nitrogen compound metabolic process	GO:0010411	1	1.47 × 10^−5^
Xyloglucan metabolic process	GO:0010411	1	0.000216
Tropine dehydrogenase activity	GO:050356	1	0.000337
Cell redox homeostasis	GO:0045454	1	0.000225
AP-5 adaptor complex	GO:0016021	1	0.000196
Ubiquinone biosynthetic process	GO:0006744	1	4.04 × 10^−5^
Sucrose metabolic process	GO:0005985	1	1.5 × 10^−5^

**Table 8 plants-12-01914-t008:** GO enrichment of upregulated transcripts in blue LED vs. white LED light of *A. membranaceus* sprouts (*p*-value < 0.05).

GO Terms	GO ID	Count	*p*-Value
Binding	GO:0005488	8	9.89 × 10^−6^
RNA-templated DNA biosynthetic process	GO:0006278	6	2.21 × 10^−6^
DNA metabolic process	GO:0006259	5	0.000327
DNA integration	GO:0015074	5	3.84 × 10^−10^
Membrane	GO:0016020	4	4.5 × 10^−5^
Proteolysis	GO:0006508	4	8.74 × 10^−5^
Plant-type secondary cell wall biogenesis	GO:0009834	4	1.51 × 10^−9^
Fatty acid biosynthetic process	GO:0006633	3	5.36 × 10^−5^
Glycosyltransferase activity	GO:0016757	3	5.12 × 10^−5^
Hydrolase activity	GO:0016787	2	0.000389
Sterol transport	GO:0015918	2	7.71 × 10^−5^
Pectin catabolic process	GO:0045490	2	4.38 × 10^−7^
Negative regulation of endopeptidase activity	GO:0010951	2	3.76 × 10^−6^
Defense response	GO:0006952	2	9.36 × 10^−11^
Integral component of membrane	GO:0016021	2	5.81 × 10^−5^
O-hydroxycinnamoyltransferase activity	GO:0050737	2	1.46 × 10^−5^
Resolution of meiotic recombination intermediates	GO:0000712	2	1.23 × 10^−7^
Cytoplasm	GO:0005737	1	0.000342
Ubiquitin-dependent protein catabolic process	GO:0006511	1	1.24 × 10^−5^
Urea cycle	GO:0000050	1	0.000185
Protein phosphorylation	GO:0006468	1	5.29 × 10^−6^
Regulation of transcription by RNA polymerase II	GO:0006357	1	0.000445
Proteolysis involved in protein catabolic process	GO:0051603	1	6.45 × 10^−5^
Signal peptide processing	GO:0006465	1	9.5 × 10^−5^
Response to oxidative stress	GO:0006979	1	0.00034
Cell wall modification	GO:0042545	1	0.000444
Auxin-activated signaling pathway	GO:0009734	1	6.54 × 10^−5^
DNA replication	GO:0006260	1	0.000185
Nucleic acid phosphodiester bond hydrolysis	GO:0090305	1	3.7 × 10^−5^
Response to light stimus	GO:0009416	1	4.13 × 10^−5^
Protein dephosphorylation	GO:0006470	1	2.99 × 10^−6^
Aldo-keto reductase (NADP) activity	GO:0004033	1	6.81 × 10^−5^
Glucose metabolic process	GO:0006006	1	8.37 × 10^−6^
Carbohydrate metabolic process	GO:0005975	1	0.000252
Organelle organization	GO:0006996	1	7.49 × 10^−6^
Nuclear-transcribed mRNA catabolic process	GO:0000184	1	0.000245
Protein glycosylation	GO:0006486	1	4.91 × 10^−7^
Cellulose microfibril organization	GO:0010215	1	6.13 × 10^−6^
Carbohydrate transmembrane transport	GO:0034219	1	4.54 × 10^−7^
Arginyl-tRNA aminoacylation	GO:0006420	1	6.82 × 10^−6^
Cellular response to phosphate starvation	GO:0016036	1	0.000438
Positive regulation of GTPase activity	GO:0043547	1	3.93 × 10^−8^
Aldehyde dehydrogenase (NAD+) activity	GO:0004029	1	8.81 × 10^−5^
Lipid transport	GO:0006869	1	2.6 × 10^−6^
Regulation of gene expression	GO:0010468	1	7.61 × 10^−6^
Regulation of stimulus	GO:0050896	1	4.24 × 10^−5^
Anthycyanin-containing compound biosynthetic process	GO:0009718	1	0.000172
Plasma membrane	GO:0005886	1	3.35 × 10^−5^
Monooxygenase activity	GO:0004497	1	5.4 × 10^−7^
Systemic acquired resistance	GO:0009627	1	1.71 × 10^−6^
Protein folding	GO:0006457	1	3.41 × 10^−12^
Threonine biosynthetic process	GO:0009088	1	5.61 × 10^−5^
Mitochondrion	GO:0005739	1	0.000105
Tricarboxylic acid cycle	GO:0006099	1	6.71 × 10^−5^
Sterol biosynthetic process	GO:0016126	1	4.89 × 10^−8^
Phosphate-containing compound metabolic process	GO:0006796	1	0.00017
Flavonoid biosynthetic process	GO:0009813	1	0.00017
Carbonyl reductase (NADPH) activity	GO:0004090	1	5.11 × 10^−8^
Group II intron splicing	GO:0000373	1	1.66 × 10^−7^
Sexual reduction	GO:0019953	1	4.97 × 10^−22^
Mitochondrial cytochrome C oxidase assembly	GO:0033617	1	1.12 × 10^−27^
mRNA destabilization	GO:0061157	1	0.000126

**Table 9 plants-12-01914-t009:** GO enrichment of downregulated transcripts in blue LED vs. white LED light of *A. membranaceus* sprouts (*p*-value < 0.05).

GO Terms	GO ID	Count	*p*-Value
DNA integration	GO:0015074	5	0.000236
Binding	GO:0005488	5	3.73 × 10^−5^
Inositol catabolic process	GO:0019310	3	4.17 × 10^−9^
RNA-templated DNA biosynthetic process	GO:0006278	3	0.000244
Translational initiation	GO:0006413	3	4.51 × 10^−14^
Transmembrane transport	GO:0055085	2	3.58 × 10^−16^
Hydrolase activity	GO:0016787	2	0.000351
Membrane	GO:0016020	2	2.23 × 10^−8^
Nucleolus	GO:0005730	2	0.000147
Glycogenin glucosyltransferase activity	GO:0008466	1	0.000124
Protein Phosphorylation	GO:0006468	1	4.76 × 10^−13^
Response to light stimulus	GO:0009416	1	1.06 × 10^−9^
Response to heat	GO:0009408	1	0.000219
Nucleic acid metabolic process	GO:0090304	1	9.48 × 10^−5^
Chloroplast organization	GO:0009658	1	4.82 × 10^−5^
Catalytic activity	GO:0003824	1	2.7 × 10^−16^
Regulation of DNA-templated transcription	GO:0006355	1	4.32 × 10^−11^
Glycine biosynthetic process, by transamination of glyoxylate	GO:0019265	1	4.23 × 10^−5^
Embryo development ending in seed dormancy	GO:0009793	1	0.000332
Naringenin 3-dioxygenase activity	GO:0045486	1	2.04 × 10^−5^
Defense response	GO:0006952	1	0.000244
Oxidative photosynthetic carbon pathway	GO:0009854	1	1.96 × 10^−24^
Citrate transport	GO:0015746	1	1.79 × 10^−8^
RNA-templated transcription	GO:0001172	1	4.84 × 10^−5^
Sterol metabolic process	GO:0016125	1	0.00045
Signal transduction	GO:0007165	1	1.3 × 10^−5^
Organic substance metabolic process	GO:0071704	1	0.000355
Protein-disulfide reductase activity	GO:0015035	1	1.02 × 10^−7^
Negative regulation of translation	GO:0017148	1	3.96 × 10^−5^
Nucleoside metabolic process	GO:0009116	1	2.45 × 10^−9^
Protein ubiquitination	GO:0016567	1	0.000405
Glucose metabolic process	GO:0006006	1	0.000261
Nucleosome assembly	GO:0006334	1	7.29 × 10^−6^
SCF-dependent proteasomal ubiquitin-dependent protein catabolic process	GO:0031146	1	0.000239
Carbon utilization	GO:0015976	1	7.75 × 10^−6^
Glutathione metabolic process	GO:0006749	1	1.9 × 10^−5^
Cellular metabolic process	GO:0044237	1	6.38 × 10^−6^
Transcription initiation at RNA polymerase II promoter	GO:0006367	1	0.000147
Oxidoreductase activity, acting on the CH-OH group of donors, NAD or NADP as acceptor	GO:0016616	1	0.000209
DNA metabolic process	GO:0006259	1	2.57 × 10^−5^
Protein stabilization	GO:0050821	1	8.95 × 10^−6^
Sucrose metabolic process	GO:0005985	1	4.04 × 10^−5^

**Table 10 plants-12-01914-t010:** GO enrichment of upregulated transcripts in red LED vs. white LED light of *A. membranaceus* sprouts (*p*-value < 0.05).

GO Terms	GO ID	Count	*p*-Value
DNA integration	GO:0015074	7	1.38 × 10^−6^
Resolution of meiotic recombination intermediates	GO:0000712	6	1.45 × 10^−7^
Proteolysis	GO:0006508	5	9.01 × 10^−7^
Flavonoid biosynthetic process	GO:0009813	4	7.29 × 10^−6^
Defense response	GO:0006952	4	4.38 × 10^−10^
Signal transduction	GO:0007165	3	4.57 × 10^−5^
Nucleic acid metabolic process	GO:0090304	3	0.000198
Xyloglucan metabolic process	GO:0010411	3	0.000246
DNA metabolic process	GO:0006259	2	3.04 × 10^−5^
Hydrolase activity	GO:0016787	2	2.09 × 10^−5^
RNA-templated DNA biosynthetic process	GO:0006278	2	2.75 × 10^−6^
DNA-templated DNA replication	GO:0006261	2	0.000408
RNA phosphodiester bond hydrolysis, endonucleoytic	GO:0090502	2	1.36 × 10^−6^
Regulation of gene expression	GO:0010468	2	0.000422
Telomere maintenance	GO:0000723	2	0.000115
Pectin catabolic process	GO:0045490	2	1.09 × 10^−5^
Negative regulation of endopeptidase activity	GO:0010951	2	1.33 × 10^−5^
DNA topological change	GO:0006265	2	0.000211
Pentose-phospho shunt, non-oxidative branch	GO:0009052	1	7.65 × 10^−5^
Negative regulation of translation	GO:0017148	1	0.000259
Transmembrane phosphate ion transport from cytosol to vacuole	GO:1905011	1	9.62 × 10^−8^
tRNA methylation	GO:0030488	1	1.32 × 10^−7^
RNA binding	GO:0003723	1	0.000337
Chromatin remodeling	GO:0006338	1	3.06 × 10^−6^
Auxin-activated signaling pathway	GO:0009734	1	6.5 × 10^−5^
Nucleic acid phosphodiester bond hydrolysis	GO:0090305	1	1.7 × 10^−6^
Dephosphorylation	GO:0016311	1	1.27 × 10^−5^
RNA phosphodiester bond hydrolysis acid bonding	GO:0090502	1	2.36 × 10^−8^
Glucose metabolic process	GO:0006006	1	8.55 × 10^−9^
Fatty acid biosynthetic process	GO:0006633	1	6.04 × 10^−5^
Carbon utilization	GO:0015976	1	2.01 × 10^−5^
Aromatic compound biosynthetic process	GO:0019438	1	1.45 × 10^−5^
Translational elongation	GO:0006414	1	2.92 × 10^−8^
Gene silencing by RNA-directed DNA methylation	GO:0080188	1	5.01 × 10^−7^
Iron ion binding	GO:0005506	1	0.000222
Carbohydrate transmembrane transport	GO:0034219	1	1.31 × 10^−9^
4-coumarate-CoA ligase activity	GO:0016207	1	0.000427
Nucleic acid binding	GO:0003676	1	1.18 × 10^−9^
Binding	GO:0005488	1	1.98 × 10^−7^
Lipid transport	GO:0006869	1	0.000201
Carbohydrate transport	GO:0008643	1	1.97 × 10^−5^
Carboxyl acid metabolic process	GO:0019752	1	3.18 × 10^−6^
Triglyceride lipase activity	GO:0004806	1	0.000147
Monooxygenase activity	GO:0004497	1	3.65 × 10^−7^
Isopentenyl diphosphate biosynthetic process, methylerylthritol 4-phosphate pathway involved in terpenoid biosynthetic process	GO:0051484	1	1.4 × 10^−11^
Protein folding	GO:0006457	1	2.87 × 10^−15^
Tricarboxylic acid cycle	GO:0006099	1	5.24 × 10^−5^
Protein ubiquitination	GO:0016567	1	0.000407
Carbonyl reductase (NADPH) activity	GO:0004090	1	6.15 × 10^−12^
Cysteine biosynthetic process from serine	GO:0006535	1	6.67 × 10^−6^
UDP-D-xylose biosynthetic process	GO:0033320	1	8.21 × 10^−5^
Sexual reproduction	GO:0019953	1	7.63 × 10^−31^
Protein phosphorylation	GO:0006468	1	0.000117
Mitochondrial cytochrome c oxidase assembly	GO:0033617	1	1.4 × 10^−27^
Protein import into nucleus	GO:0006606	1	1.43 × 10^−6^
Integral component of membrane	GO:0016021	1	3.8 × 10^−6^
Cell redox homeostasis	GO:0045454	1	0.000211
Ubiquinone biosynthetic process	GO:0006744	1	8.1 × 10^−5^
Stress granule assembly	GO:0034063	1	7.95 × 10^−5^
mRNA destabilization	GO:0061157	1	9.94 × 10^−5^

**Table 11 plants-12-01914-t011:** GO enrichment of downregulated expressed transcripts in red LED vs. white LED light of *A. membranaceus* sprouts (*p*-value < 0.05).

GO Terms	GO ID	Count	*p*-Value
DNA integration	GO:0015074	6	6.93 × 10^−5^
Nucleic acid metabolic process	GO:0090304	3	1.77 × 10^−5^
Binding	GO:0005488	3	6.91 × 10^−7^
Phosphorylation	GO:0016310	2	0.000381
Diterpenoid biosynthetic process	GO:0016102	2	2.09 × 10^−7^
Hydrolase activity	GO:0016787	2	0.000444
Membrane	GO:0016020	2	2 × 10^−8^
Nucleolus	GO:0005730	2	5.14 × 10^−5^
RNA-templated DNA biosynthetic process	GO:0006278	1	1.99 × 10^−5^
Protein phosphorylation	GO:0006468	1	1.43 × 10^−5^
response to red light	GO:0010114	1	8.83 × 10^−5^
response to heat	GO:0009408	1	0.000214
Chloroplast organization	GO:0009658	1	2.52 × 10^−6^
Plasma membrane	GO:0005886	1	0.000189
Acyltransferase activity	GO:0016787	1	0.000432
Cation transmembrane transport	GO:0098655	1	0.000137
Photosynthetic electron transport in photosystem I	GO:0009733	1	2.02 × 10^−7^
Proteolysis	GO:0006508	1	0.000415
Mitochondrion	GO:0005739	1	2.79 × 10^−5^
Xylan catabolic process	GO:0045493	1	1.2 × 10^−6^
Coenzyme A biosynthetic process	GO:0015937	1	0.000408
Cellular amino acid biosynthetic process	GO:0008652	1	0.000421
Peptydyl-serine phosphorylation	GO:0018105	1	0.000298
RNA-templated transcription	GO:0005730	1	8.96 × 10^−8^
Signal transduction	GO:0007165	1	8.18 × 10^−5^
Protein dimerization activity	GO:0046983	1	4.92 × 10^−7^
Sucrose biosynthetic process	GO:0005986	1	7.16 × 10^−7^
Exonucleolytic trimming to generate mature 3′-end of 5.8S rRNA from tricistronic rRNA transcript (SSU-rRNA, 5.8S rRNA, LSU-rRNA)	GO:0000467	1	8.71 × 10^−5^
4 iron, 4 sulfur cluster binding	GO:0051539	1	0.000264
Inositol catabolic process	GO:0019310	1	9.2 × 10^−6^
Calcium ion binding	GO:0005509	1	6.83 × 10^−5^
Protein ADP-ribosylation	GO:0006471	1	5.45 × 10^−21^
Protein-disulfide reductase activity	GO:0015035	1	1.36 × 10^−6^
Chloroplast rRNA processing	GO:1901259	1	2.6 × 10^−6^
UDP-D-xylose biosynthetic process	GO:0033320	1	4.04 × 10^−6^
Microtubule-based movement	GO:0007018	1	9.91 × 10^−7^
Protein ubiquitination	GO:0016567	1	0.000136
SCF-dependent proteasomal ubiquitin-dependent protein catabolic process	GO:0031146	1	0.000461
Lipid transport	GO:0006869	1	0.000368
Autophagosome assembly	GO:0000045	1	2.22 × 10^−6^
DNA metabolic process	GO:0006259	1	3.57 × 10^−23^
Fatty acid biosynthetic process	GO:0006633	1	1.96 × 10^−5^
Mitochondrial mRNA modification	GO:0080156	1	0.000184

**Table 12 plants-12-01914-t012:** Primer sequence for reference genes to qPCR analysis.

Reference Gene ID	Primer Sequence (5′→3′)
Gene_079740T	forward: TGACGCCTGATGCTGCATAT
	reverse: AAGGTGGCGGTAGTAGTCCT
Gene_119030T	forward: TGGCAACCATTTTGCTGA
	reverse: TCCTTCCATGCAAGGCAACA
Gene_174710T	forward: AGGAAGAGATAGTGGCGATGA
	reverse: TGATCTCCAAGGCGATGCAA
Gene_334410T	forward: CCCGTCGCACAACTAGAGAT
	reverse: GAACGCCTTGCTGCATCTTG
Gene_447300T	forward: ATCCAACGCCTCAAACACTC
	reverse: AGAGTGCACCCATGTTGTTG
Gene_054270T	forward: GGAGCAATTGGATGAGCCCT
	reverse: ACCAGCACCACGAATATTCCA
Gene_102000T	forward: AGCTCCATGCCATCACTAGC
	reverse: AGTGTTGTTGCTCCGGAGTT
Gene_277270T	forward: GAGCCCTCTGCAACCAACTC
	reverse: GCAGAGTTCACCTGGTGTGT
Gene_344410T	forward: CCTGATGCAAACATGTTCCCC
	reverse: TCATTCATGGCAGTTGCACC
Gene_377000T	forward: AATCGACGGGCAAATGGAGA
	reverse: GTGAATTTCTGTGTCGGCGC

## Data Availability

The data presented in this study are contained within the article.

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
