# Peer review of "Cellular Morphology and Transcriptome Comparative Analysis of *Astragalus membranaceus* Bunge Sprouts Cultured In Vitro under Different LED Light"

_plants, 2023, doi:10.3390/plants12091914_

Round 1

Reviewer 1 Report (Previous Reviewer 2)

I have reviewed this manuscript. I have no more comments. Thanks.

Author Response

Dear Reviewer1

Thanks much.

Best regards,

Prof. Eun Soo Seong

Reviewer 2 Report (New Reviewer)

The manuscript entitled “Cellular Morphology and Transcriptome Comparative analysis of Astragalus membranaceus Bunge. Sprouts Cultured in vitro under different LED light” by Seo et al. aimed to study the utility of A. membranaceus as a sprout crop using LED light sources. The study showed that three different LED light sources have differentially affected gene expression levels on the plants. These results may provide insights into the establishment of a flavonoid biosynthesis regulation model and the cultivation of A. membranaceus. 

Overall, this study is interesting, and the conclusions are appropriate, and supported by the data. I recommend accepting it.

Author Response

Dear Reviewer2

Thanks much.

Best regards,

Prof. Eun Soo Seong

Reviewer 3 Report (New Reviewer)

Article Cellular morphology and transcriptome comparative analysis of Astragalus membranaceus Bunge. sprouts cultured in vitro under different LED lights by authors Ji Won Seo, Jae Geun Lee , Ji Hye Yoo , Jung Dae Lim, Ik-Young Choi, Myong Jo Kim, Chang Yeon Yu, Eun Soo Seong is a comprehensive study on microgreens from a plant with a number of valuable secondary metabolites. Despite the use of various approaches, the study leaves a double impression and requires some simple revision and very careful proofreading of the manuscript.

So even the beginning of the abstract is bewildering. The authors claim that astragalus consists of components. Probably even so there were secondary metabolites included in (?) leaves, raw materials, roots?

Further, it is not clear why the sprouts were studied, this should be clearly described, indicating microgreens as a potential functional nutrition and its perspective from the point of view of nutraceuticals.

The introduction should accordingly be expanded in this aspect with appropriate references, especially considering that not all of the substances mentioned are useful and safe.

As for the study itself, if routine molecular methods are performed tolerably and analyzed according to today's approximate approaches by analogy, then the routine method of analyzing surfaces using scanning microscopy is surprising. Probably the authors are not familiar with the terminology. So they call the cells of the epidermis of the adaxial surface of the leaf some kind of leaf cells, the approach of the authors to the name of the cells of the epidermis of the stem is no less interesting. Obviously, such data look rather strange without data on the abaxial side, on which the stomata are known to be located. It is not clear how many leaves of what age the authors analyzed in each case, nor is it clear why such a method should be used when it is visible under a conventional microscope. But the number of stomata could explain a lot, since it is connected with respiration (which provides each metabolic process with energy and photosynthesis, since the balance of carbon dioxide and oxygen is associated with the development of aerenchyma and the availability of these compounds.

There are questions about the method of fixation. For plants, not cocadylate, but sodium phosphate buffer is usually used. If the authors suggest this method and do not equalize the osmotic pressure with sucrose, this should be reflected or referenced.

It is not clear what the inscription chlorophyll a means in the histograms of the spectra.

The inscriptions in figures 3-4-5 are very illegible.

As for the isolation, it remains unclear from what the nucleic acids were isolated - the authors used the term sprouts, if it is a conditionally green part, then this is one thing, if together with the roots, then it is completely different and it is not clear how correct this is, since the synthesis of metabolites in these parts is fundamentally different.

Near the plants, it is necessary to show the dimension (ruler).

It is required to answer these questions before a detailed consideration of the nuances.

However, I don't think it will take long.

Author Response

Response to reviewer 3

Thank you for your review.
Among the pointed out contents, those that can be corrected have been corrected and marked in the text in red.

Q1. So even the beginning of the abstract is bewildering. The authors claim that astragalus consists of components. Probably even so there were secondary metabolites included in (?) leaves, raw materials, roots?

R1. The reviewer said it was bewildered from the beginning of the abstract, and this is a well-known fact in Astragalus membranaceus plants. In addition, as a basis for it, I have already added some citations on the secondary metabolites of Astragalus membranaceus. No one can deny this fact.

Q2. Further, it is not clear why the sprouts were studied, this should be clearly described, indicating microgreens as a potential functional nutrition and its perspective from the point of view of nutraceuticals.

R2. Astragalus is a medicinal plant. Its value is fully explained in the introduction. In addition, the research at the sprout vegetable level was conducted to provide basic data for use in the plant factory system, and this part is well explained in the introduction. The nutritional value will be discussed and evaluated in the next paper. The main content of this study is the comparative analysis of the artificial light source effect and the transcriptome of A. membranaceus sprout.

Q3. The introduction should accordingly be expanded in this aspect with appropriate references, especially considering that not all of the substances mentioned are useful and safe.

R3. It is not correct to extend the description to the stability of the substances mentioned. The points pointed out by the reviewer should be addressed in the thesis on food safety evaluation. No other reviewers have pointed out that this part should be expanded.

Q4. As for the study itself, if routine molecular methods are performed tolerably and analyzed according to today's approximate approaches by analogy, then the routine method of analyzing surfaces using scanning microscopy is surprising. Probably the authors are not familiar with the terminology. So they call the cells of the epidermis of the adaxial surface of the leaf some kind of leaf cells, the approach of the authors to the name of the cells of the epidermis of the stem is no less interesting. Obviously, such data look rather strange without data on the abaxial side, on which the stomata are known to be located. It is not clear how many leaves of what age the authors analyzed in each case, nor is it clear why such a method should be used when it is visible under a conventional microscope. But the number of stomata could explain a lot, since it is connected with respiration (which provides each metabolic process with energy and photosynthesis, since the balance of carbon dioxide and oxygen is associated with the development of aerenchyma and the availability of these compounds.

R4. This data is not meant to measure stomata. When viewed with the naked eye, each artificial light source showed a difference in biomass, so only the difference in plant cell size was measured using SEM. Also, the terminology related here was requested by the reviewers of Plants to be modified as the current terminology during the first submission. With SEM, only the cell size can be observed regardless of the stomata shape. The measured leaf samples were observed by taking 3 leaves with the 4th from the top. This content has been modified in materials and methods of text.

Q5. There are questions about the method of fixation. For plants, not cocadylate, but sodium phosphate buffer is usually used. If the authors suggest this method and do not equalize the osmotic pressure with sucrose, this should be reflected or referenced.

R5. When preparing samples for SEM measurement, cocadylate buffer is sometimes used. The references [31] are attached.

Q6. It is not clear what the inscription chlorophyll a means in the histograms of the spectra.

A6. What are you not sure about? I don't understand. This is the phrase that came out of the machine when measured with a device [PG200N illuminometer (United Power Research Technology Co., Zhunan Township, Taiwan)]. It's not the text I typed. The picture and word that came out of in the device. See Measuring Instruments in Materials and Methods. There was no problem when it was published in other academic journals.

Q7. The inscriptions in figures 3-4-5 are very illegible.

A7. What is the “inscription” you asked about? Exact word...is it right? If you are talking about a phrase that corresponds to a picture, that phrase is no problem to interpret the picture. Other reviewers did not point this out either.

Q8. As for the isolation, it remains unclear from what the nucleic acids were isolated - the authors used the term sprouts, if it is a conditionally green part, then this is one thing, if together with the roots, then it is completely different and it is not clear how correct this is, since the synthesis of metabolites in these parts is fundamentally different.

A8. Total RNA was isolated. What is the problem? There is no problem in RNA isolation.

Q9. Near the plants, it is necessary to show the dimension (ruler).

A9. Table 1 shows the measured results again. See Table1.

This manuscript is a resubmission of an earlier submission. The following is a list of the peer review reports and author responses from that submission.

Round 1

Reviewer 1 Report

The present manuscript is focused on the in vitro grown legume plant Astragalus membranaceus and the effect of light-emitting diodes (LED) light sources (white, red, blue) on cell size and transcriptome. The cell morphology analysis revealed that leaf cell size was largest in plants exposed to red light, and the thickest stem cells were observed in plants exposed to white light. RNA sequencing resulted in pooling out differentially expressed up-regulated and down-regulated genes among the light sources. The transcriptome data were subjected to Gene Ontology (GO) enrichment analysis, which highlighted different categories: cellular process, catalytic activity, cellular process, which demonstrated that different LED light sources can differentially influence the transcriptome expression pattern of A. membranaceus sprouts. The GO enriched in the white LED vs. red LED comparison was associated with flavonoid biosynthesis, providing a basis for establishing a strategy for cultivation of high-functional Astragalus sprouts.

The manuscript presents an interesting dataset which can be extrapolated to other legumes, too. The results are presented clearly and discussed appropriately. In my opinion, the data would be of interest for a broad audience.

Comments:

Abstract – it is better to be shorten. Exact cell size and RNA seq reads, GO category (number and number of items), are not necessarily needed to be included in.

Page 1, Lines 33-34 – the variants are repeated again, and this can be avoided by link with the previous sentence.

Page 2, Line 83 – rather than “still far from incomplete”, probably you mean “still far from complete”.

Page 3, Line 114 – are these “cross sections”?

Page 4, Line 140 – the name of the program is mentioned in Methods, so no need to be included here.

Figures 4 and 5 – higher resolution for the text is required.

Table 6 – please, correct “Negatice regulation”, and in Table 7 – “Nocleoside metabolic process”; Table 8 – “Mitichndrion”; Table 9 – “Transcrition initiation at RNA polymerase II promoter”

Page 15, Line 281 – please, explain the abbreviation BA

Page 15, Lines 295-299 – please, indicate the literature reference. Do you really mean white LED light or red?

Page 16, Line 343 – please, re-write “those plant high cultivated” since it is not clear.

Page 16, Lines 358-360 – Why the conclusion is only about white LED light?

Author Response

Response to reviewer 1

* Abstract was shorten depending on reviewer opinion.

* As your request, I have marked in red the revised parts in the manuscript.

Q1: Page 1, Lines 33-34 – the variants are repeated again, and this can be avoided by link with the previous sentence.

A1: The part was deleted.

Q2: Page 2, Line 83 – rather than “still far from incomplete”, probably you mean “still far from complete”.

A2: I revised to “still far from complete”.

Q3: Page 3, Line 114 – are these “cross sections”?

A3: “by cross sections” was inserted in this sentence.

Q4: Page 4, Line 140 – the name of the program is mentioned in Methods, so no need to be included here.

A4: The name of the program was deleted.

Q5: Figures 4 and 5 – higher resolution for the text is required.

A5: It was resolved.

Q6: Table 6 – please, correct “Negatice regulation”

A6: I revised to “Negative regulation”

Q7: Table 7 – “Nocleoside metabolic process”

A7: I revised to “Nucleoside metabolic process.”

Q8: Table 8 – “Mitichndrion”

A8: I revised to “Mitochondrion.”

Q9: Table 9 – “Transcrition initiation at RNA polymerase II promoter”

A10: I revised to “Transcription initiation at RNA polymerase II promoter.”

Q10: Page 15, Line 281 – please, explain the abbreviation BA

A10: Full name was inserted.

Q11: Page 15, Lines 295-299 – please, indicate the literature reference. Do you really mean white LED light or red?

A11: Reference was inserted.

Q12: Page 16, Line 343 – please, re-write “those plant high cultivated” since it is not clear.

A12: I revised to “plant cultivated”.

Q13: Page 16, Lines 358-360 – Why the conclusion is only about white LED light?

A13: I inserted in final sentence “In addition, it is suggested that the light condition for improving the biomass of A. membranaceus sprouts require appropriate use of white-light and red-light sources.”

Reviewer 2 Report

In the manuscript from Ji Won Seo, the authors reported that cellular morphology and transcriptome show changed in A. membranaceus Bunge. Sprouts when treated by different LED lights in vitro.  The Authors keep A. membranaceus (b) grown in vitro cultured under 3 types of LEDs lights for 6 weeks, then do scanning electron microscope observe the cell morphology. They found that red LED light exposure can increase the cell size of A. membranaceus. The RNA-seq results suggested that some genes are differently regulated in these plants treated by different wavelength light, which are involved in different molecular functions, cellular components and biological processes.

This manuscript is well written and easy to follow. However, the results are not clearly presented, and the research designs are lack of rationality. Even though a lot of RNA-seq works were done, the authors only describe the results superficially and they didn’t mine deeply the mechanism under these data. The following are some my comments:

Major concerns:

1.       In the section 2.1, the authors only describe that the cellular morphologies of A. membranaceus were affected by the different LED light exposure. I want to ask that does treatment by different LED lights have impacts on the biomass of A. membranaceus? Is the different cell morphologies have an impact on biomass? Do the authors find the expression of genes related to cell morphology were affected by different LED exposure?

2.        A. membranaceus has been used as a medicinal plant. The secondary metabolites such as formononetin, saponins and flavonoids are the major functional component. Do the different LED light treatments have an impact on the secondary metabolites profile of A. membranaceus? Do the author analysis the genes expression that are involved in secondary metabolite production are affected by different LED lights?

Minor concerns:

1.       Line 117, “stem cell thickness” should be “stem cell width”.

2.        Figure5. The font size in the image is too small to see clearly.

Author Response

Response to reviewer 2

* As your request, I have marked in red the revised parts in the manuscript.

Q1: In the section 2.1, the authors only describe that the cellular morphologies of A. membranaceus were affected by the different LED light exposure. I want to ask that does treatment by different LED lights have impacts on the biomass of A. membranaceus? Is the different cell morphologies have an impact on biomass? Do the authors find the expression of genes related to cell morphology were affected by different LED exposure?

A1: This manuscript is a study to suggest the industrialization possibility of using the medicinal plant sprout A. membranaceus as a vegetable. It was a very difficult task to compare the entire transcriptome by comparing 2 treatments for 3 light source conditions. Therefore, after the publication of this paper, the author will of course continue to study the functional analysis of genes related to cell morphology and biomass of A. membranaceus. As already stated in the manuscript, the leaf cell area was the largest under red light, and the stem cell area was the largest under white light. Therefore, it can be inferred that genes up-regulated in white and red light are related to biomass.

Q2: A. membranaceus has been used as a medicinal plant. The secondary metabolites such as formononetin, saponins and flavonoids are the major functional component. Do the different LED light treatments have an impact on the secondary metabolites profile of A. membranaceus? Do the author analysis the genes expression that are involved in secondary metabolite production are affected by different LED lights?

A2: Currently, I am researching the function of genes related to formononetin and calycosin content in A. membranaceus. Details related to this will be submitted in the next paper. Again, there is little research on the use of medicinal plants as sprouts, so please focus your review on this. What the reviewers pointed out is a topic currently being researched based on the transcriptome information submitted to this manuscript.

Q3: Line 117, “stem cell thickness” should be “stem cell width”.

A3: I revised to “stem cell width”.

Q4: Figure5. The font size in the image is too small to see clearly.

A4: I resolved.

Round 2

Reviewer 2 Report

I have no more comments..